# The Acute Effect of Warm-Up with Cold Water Immersion upon Calf Raise Performance, Muscle Tension, and Oxygen Saturation

**DOI:** 10.3390/jfmk10030328

**Published:** 2025-08-27

**Authors:** Roland van den Tillaar, Patrick Lunde, Milosz Mielniczek

**Affiliations:** Department of Sports Sciences and Physical Education, Nord University, 7600 Levanger, Norway; patricklunde@live.no (P.L.); milosz.msg@gmail.com (M.M.)

**Keywords:** resistance training, RPE, muscle oxygenation, CWI

## Abstract

**Objectives:** This study investigated the acute effects of pre-exercise cold-water immersion (CWI) on performance, muscle oxygen saturation, and mechanical muscle tension during calf raise training. **Method:** Twenty-four trained individuals (5 females, 19 males) were randomly assigned to either a CWI group (5 min at 10 ± 1 °C) or a non-CWI group (no intervention). Both groups performed three sets of standing calf raises to failure using a standardized protocol. Load lifted, repetitions, and rate of perceived exertion (RPE) were recorded. Muscle oxygenation (SmO_2_, total hemoglobin) and mechanical muscle properties (frequency and stiffness) were measured before and after each set. **Results:** The CWI group showed a significantly greater increase in barbell load from set 1 to set 2 compared to the non-CWI group (from 94.5 ± 18.1 kg to 98.0 ± 18.7 kg, *p* < 0.01). Repetitions decreased and RPE increased across sets in both groups. The non-CWI group exhibited earlier increases in muscle stiffness and frequency, whereas these responses were delayed in the CWI group. Gastrocnemius SmO_2_ increased during the protocol in the non-CWI group only. Total hemoglobin change was greater in the CWI group in set 1. **Conclusions:** These findings suggest that pre-exercise CWI may acutely enhance performance and delay neuromuscular fatigue without negatively affecting perceived effort.

## 1. Introduction

Cold-water immersion (CWI) is widely applied in sports and exercise science as a recovery modality, owing to its purported physiological benefits such as reduced muscle soreness, enhanced muscle recovery, and improved subsequent performance [1,2]. These effects are thought to be mediated by mechanisms including vasoconstriction-induced reductions in muscle blood flow [3], alterations in muscle oxygenation [2], and the modulation of inflammatory responses [4,5]. Despite these benefits, recent research has indicated that regular post-exercise CWI may impair long-term resistance training adaptations, particularly by attenuating muscle hypertrophy and vascular remodeling [6,7,8]. These conflicting findings underscore the complexity of CWI’s effects on muscle function and adaptation and highlight the need for further investigation into both its acute and chronic impacts. In recent years, interest has also grown around inter-set cooling strategies, particularly distal cooling of the palms and soles, which has been proposed to enhance total resistance training volume and muscular activation [9,10,11,12,13]. For instance, Kwon et al. [11] reported that applying palm cooling for 2.5 min between sets significantly increased training volume compared to both hand warming and neutral conditions. However, McMahon et al. [14], using only one minute of palm cooling, did not observe similar performance benefits. Similarly, cooling of the soles during rest intervals has been linked to improved outcomes, such as greater repetitions and increased EMG activity during leg press exercises [9,13]. Although the precise physiological mechanisms remain unclear, inter-set cooling has been associated with elevated EMG responses in target muscles, reduced ratings of perceived exertion (RPE), and heightened arousal levels compared to control conditions [9,11,12,13]. Beyond these neural effects, enhanced peripheral blood flow and heat dissipation [15,16,17] through thermoregulatory pathways may also contribute to the observed performance gains. Additionally, CWI has been shown to reduce muscle oxygen consumption (mVO_2_), reflecting a lowered metabolic rate within the muscle tissue post-immersion. This may influence subsequent performance and neuromuscular activation by altering both oxygen availability and energy turnover during exercise [18].

While most studies to date have focused on post-exercise or inter-set CWI, the potential of pre-exercise CWI remains underexplored. Pre-cooling strategies such as CWI have primarily been studied in the context of endurance performance, particularly under heat stress, where they have been shown to enhance exercise capacity [19]. However, the effects of pre-exercise cooling on resistance training performance are still poorly understood. The physiological responses to cold exposure prior to exercise may differ substantially from post-exercise and inter-set applications. For instance, pre-exercise cooling may impair neuromuscular function by reducing nerve conduction velocity, muscle contractility, and metabolic efficiency, potentially diminishing force output during resistance exercise [20,21]. Furthermore, alterations in muscle blood flow and oxygenation resulting from pre-exercise cold exposure may influence acute performance outcomes [2,22].

However, to the best of our knowledge, no studies have investigated the acute effects of full-body CWI applied prior to resistance exercise. Given the physiological changes associated with cold exposure, such as altered muscle oxygenation, neuromuscular activation, and thermoregulation, it is plausible that pre-exercise CWI may influence subsequent performance. Therefore, the primary objective of this study is to evaluate the acute effects of full-body pre-exercise CWI on calf raise performance, muscle oxygenation (SmO_2_ and tHb), and muscle tension (via myotonometry) in resistance-trained individuals. Participants were randomly assigned to either a CWI or control condition, and performance variables were assessed across three consecutive sets. This experimental design allows for a direct comparison of physiological and performance outcomes between groups, thereby providing operational insight into whether pre-exercise CWI facilitates or impairs strength training capacity. By addressing this unexplored application of CWI, the study aims to offer evidence-based guidance for practitioners considering pre-cooling as part of a warm-up routine.

## 2. Materials and Methods

The present study was part of a longitudinal study on the effect of CWI as a warm-up before strength training in which a non-CWI and CWI group conducted an intervention. In that study, the longitudinal effects of CWI are investigated. However, it is not known what acute happens during one training session upon strength training performance and possible involved processes (oxygen saturation, total hemoglobin, and muscle tension). Therefore, in the middle of the intervention period a training session was used for both groups to study the acute effects upon these parameters to obtain more information about possible mechanisms that occur after CWI as warm-up before strength training. Thereby, a between-subject design was used. Calf raises in a Smith machine were used as this exercise is easy to measure the muscles and not many train calf raises regularly and therefore it is easier to recruit and control the participants in their training.

### 2.1. Participants

For this study, 24 participants (5 women, 19 men) were recruited to participate in the study (age: 23.3 ± 6.3 years, height: 1.80 ± 0.09 m, body mass: 76.0 ± 11.4 kg) as this number of subjects was enough for investigating the longitudinal effects of CWI as a warm-up. Participants were recruited through personal networks at the university and via posters. The group was equally divided into a CWI and a non-CWI group. Inclusion criteria were as follows: (a) no injury or other physical illness that could reduce maximal performance. Additionally, participants were instructed not to consume any alcohol, or to conduct any training or hard physical activity for the calf <48 h prior to the test session. Before participation, written consent was obtained. The study approved by the Regional Committees for Medical and Health Research Ethics (project number: 847104) following the most recent revision of the Helsinki Declaration.

Participants were randomly assigned to groups using an online generator (Random.org/lists, Dublin, Ireland). Although the test leader and participants were not blinded to group allocation, the sessions were supervised by the same individuals, and standardized protocols for load progression and failure criteria were applied.

### 2.2. Procedure

First four weeks of training was conducted to familiarize the subjects with the exercise and the intervention group with the routine. All sessions during the four-week period were supervised and followed an identical, standardized protocol across participants as part of a larger longitudinal study. The CWI group started the session with sitting for 5 min in an ice bath (Avantopool Kide, Helsinki, Finland) that had a temperature of 10 ± 1 degrees with their whole body until their armpits were under water and their arms. The non-CWI group started straight with the calf raise exercise after the attachment of a near-infrared spectroscopy device to the legs. After the CWI, the participants had 5 min rest in which the near-infrared spectroscopy device (Moxy, Fortiori Design LLC, Hutschinson, MN, USA) was attached to the lateral gastrocnemius of the left leg and the soleus of the right leg (Figure 1), when they were laying supine on their belly on the floor on a mat. After attaching the Moxy, the participant performed their calf raises, while standing with the balls of their feet on a step and shoulders under the load of a Smith machine, (GymSport AS, Trondheim, Norway). They lowered their heels as far as possible, followed with raising them as high as possible (Figure 1). The prescribed load, which they could lift between 15 and 20 times during the previous session, was used as the load for the first set. When the participant could lift between 21 and 24 times, the load in the next set was increased by 2.5 kg; when the number of repetitions was more than 25, the load was increased by 5 kg in the next set. When a participant could only perform fewer than 15 repetitions, the load for the next set was decreased by 2.5 kg. Three sets with full exhaustion (full exhaustion is defined as the point at which the load cannot be lifted to the required height for two consecutive repetitions) in each set were performed with approximately 5 min rest between each set. The participants were allowed to use their own tempo in the calf raises to avoid an extra constraint and the set was finished when the participant could not lift the load to the height as in the tent repetitions two times in a row.

### 2.3. Measurements

After each set, and before the first set, the rate of perceived exertion (RPE) on a Borg scale (0–10) was registered [23]. Furthermore, the number of approved repetitions and the lifted load in each set were registered. A repetition was counted as successful when the participant went all the way down within their mobility limits, which vary from person to person, and up above approximately 30 percent. However, participants were allowed to go higher if their range of motion permitted it.

Just before the start of each set, the Moxy monitor (Fortiori Design LLC, Hutschinson, MN, USA) was started to measure muscle oxygen saturation (%) and total hemoglobin (g/dL) during the sets and stopped straight after finishing the set. The oxygen saturation and total hemoglobin at the start and the end, together with the minimal oxygen saturation during each set, were taken for further analysis together with the change in oxygen saturation between the start, minimal, and end of each set and change in total hemoglobin between the start and end to investigate if this varies over the sets between groups.

Muscle tension was measured at the medial Gastrocnemius of the right leg and soleus of the left leg when the participants were placed in the prone position on the mat on the floor. The measurement point was marked at the largest cross-section of the muscle belly. A hand-held digital palpation device, MyotonPRO (Myoton AS, Tallinn, Estonia), was used to measure the superficial muscle parameters of these two muscles. An initial force of 0.18 N was exerted by using a standard 3 mm-diameter probe placed perpendicularly to the skin’s surface, directly above the muscle. This was followed by an induced muscle deformation due to the additional mechanical force of 0.4 N being applied to the subcutaneous tissue for 15 milliseconds. The resultant damped natural oscillations caused by the viscoelastic properties of the soft tissue were recorded using a built-in accelerometer at a sampling rate of 3200 Hz [24]. Frequency (Hz), dynamic stiffness (N/m), decrement, relaxation time, and creep were measured with the device and used for further analysis.

### 2.4. Statistical Analysis

Normality of data distribution was assessed and confirmed using the Shapiro–Wilk test. To investigate the acute effect of CWI, a 2 (group: CWI vs. non-CWI) × 3 (sets: 1–3 repeated measures) analysis of variance (ANOVA) was conducted to assess the effects over the different sets on lifting loads, muscle tension parameters, RPE, oxygen saturation, and total hemoglobin. When significant differences were found, post hoc comparison was conducted with a Holm–Bonferroni correction. Furthermore, one-way ANOVAs per group were conducted to investigate the development per group. Statistical significance was set at an alpha level of *p* < 0.05. Effect size was evaluated with (Eta partial squared), where 0.01 < η^2^ < 0.06 constitutes a small effect, 0.06 < η^2^ < 0.14 a medium one, and η^2^ > 0.14 a large effect [24]. All analyses were conducted in JASP (version 0.18.1, Amsterdam, The Netherlands). A power analysis using G*Power v3.1.9.6 [25], based on an assumed effect size of 0.93 from prior CWI research [26] determined that 12 participants in total are sufficient to achieve 80% power (β = 0.8) at a significance level of α = 0.05. Although the sample included relatively few female participants, no apparent sex-related differences were observed between groups, and subgroup analysis was therefore not pursued.

## 3. Results

The load, number of repetitions, and RPE were significantly affected by the sets (F_(2,44)_ ≥ 18, *p* < 0.001, η^2^ ≥ 0.45). Furthermore, a significant group × set interaction effect was found for load and repetitions (F_(2,44)_ ≥ 6.6, *p* ≤ 0.003, η^2^ ≥ 0.23), with no significant group effects in any of these three variables (F_(1,22)_ ≤ 2.7, *p* ≥ 0.116, η^2^ ≤ 0.11). Post hoc comparison revealed that RPE increased from time to time in both groups. In addition, repetitions decreased from set 1 to 2 in both groups, while the lifting load only increased significantly in the CWI group from set 1 to 2 due to number of repetitions over 20 in set 1 (Table 1).

Minimal muscle oxygen saturation at the start and over the three sets was significantly affected for the gastrocnemius (F_(2,44)_ ≥ 6.4, *p* ≤ 0.004, η^2^ ≥ 0.26), but not for the soleus (F_(2,44)_ ≤ 2.3, *p* ≥ 0.122, η^2^ ≤ 0.09). Furthermore, a significant group × set interaction effect was found for oxygen saturation level at the end of the set for the gastrocnemius and at the start of the set for the soleus (F_(2,44)_ ≥ 3.5, *p* ≤ 0.04, η^2^ ≥ 0.14), with no significant group effects in any of the oxygen saturation levels (F_(1,22)_ ≤ 1.1, *p* ≥ 0.287, η^2^ ≤ 0.05). Post hoc comparison showed that oxygen saturation in gastrocnemius was significantly increased at the start of sets 2 and 3 in the non-CWI group compared with the start of set 1, while the minimal saturation also was increased from set 1 with 3 for this group.

Total hemoglobin was also significantly affected at the start and end by the sets for both muscles (F_(2,44)_ ≥ 12.4, *p* ≤ 0.001, η^2^ ≥ 0.36), but no significant interaction (F_(2,44)_ ≤ 2.7, *p* ≥ 0.077, η^2^ ≤ 0.11) or group effects (F_(1,22)_ ≤ 0.61, *p* ≥ 0.44, η^2^ ≤ 0.03) were found. Post hoc comparison showed that total hemoglobin at the start and end of the first set were significantly lower than in the other two sets (Figure 2).

When calculating the change in oxygen saturation and total hemoglobin from start to end between sets and groups, a significant effect was found for change from the start to minimal and end of oxygen saturation for the gastrocnemius and total hemoglobin of the soleus (F_(2,44)_ ≥ 4.6, *p* ≤ 0.016, η^2^ ≥ 0.17). No significant interaction effect was found, but for the soleus a near significance with moderate effect size for the change in oxygen saturation from the start to minimal and end of the sets was found (F_(2,44)_ = 2.99–3.08, *p* = 0.056–0.06, η^2^ = 0.12). No significant group effects (F_(1,22)_ ≤ 2.2, *p* ≥ 0.152, η^2^ ≤ 0.09) were found. Post hoc comparison revealed that change in oxygen saturation was significantly lower in the first set compared with sets 2 and 3 in gastrocnemius for the start to end for both groups and from the start to minimal saturation in the non-CWI group, while change in oxygen saturation of soleus increased significantly in non-CWI group from set 1 with set 3. In addition, the change in total hemoglobin in soleus was significantly higher in set 1 for the ice-bath group compared with sets 2 and 3 and with the non-CWI group in set 1 (Figure 3).

A significant effect of the sets was also found for all muscle tension variables (F_(2,44)_ ≥ 5.7, *p* ≤ 0.002, η^2^ ≥ 0.21). Furthermore, only a significant interaction effect for the decrement of the gastrocnemius (F_(2,44)_ = 3.2, *p* = 0.028, η^2^ = 0.13) was found, with no other significant interaction and group effects of any of the other muscle tension variables (F_(2,44)_ ≤ 1.6, *p* ≥ 0.208, η^2^ ≤ 0.07). Post hoc comparison revealed that muscle frequency and stiffness increased significantly from before the test to after set 1 for the non-CWI group for both muscles, while the frequency and stiffness for the CWI group did only increase significantly after set 2 for both muscles (Figure 4). Decrement, relaxation time, and creep all significantly decreased from before training to after the first set for the non-CWI group for both muscles, while for the CWI group only a significant decrease was found after set 2 for most of these variables (Figure 4). However, from sets 2 to 3 only the CWI group for the relaxation time and creep significantly decreased again for the gastrocnemius muscles (Figure 4).

## 4. Discussion

This study investigated the acute effects of pre-exercise cold water immersion (CWI) on calf raise performance, muscle tension, and muscle oxygenation in resistance-trained individuals. The main finding was that the CWI group performed more repetitions in the first set and, consequently, lifted more load in the second set compared to the non-CWI group. This performance enhancement occurred despite no group differences in overall perceived exertion. Accompanying this were distinct physiological patterns. These included greater changes in total hemoglobin in the soleus muscle during the first set in the CWI group, as well as delayed changes in both oxygen saturation and muscle tension compared to the non-CWI group.

The enhanced performance in the first set in the CWI group may be explained by the acute physiological effects of cold exposure on neuromuscular function and vascular dynamics. Cold water immersion has been shown to transiently reduce skin and superficial muscle temperature. This cooling effect may lead to a short-term attenuation of fatigue development through mechanisms such as decreased nerve conduction velocity, reduced inflammation, or altered recruitment patterns [2]. This could preserve muscular force-generating capacity early in the session, explaining the higher repetition count and subsequently increased load in the second set. Previous research has demonstrated that cold-water immersion at both 10 °C and 15 °C significantly reduces muscle oxygen consumption (mVO_2_), indicating a suppression of metabolic activity post-immersion [18]. This aligns with the present study’s findings. The ice bath group exhibited attenuated changes in oxygen saturation and delayed muscle tension responses, possibly due to decreased muscle metabolism induced by the cold exposure.

One particularly relevant mechanism involves the significantly greater increase in total hemoglobin (tHb) in the soleus muscle during set 1 in the CWI group. While baseline values were slightly lower (though not significantly), the larger increase (Figure 2 and Figure 3) suggests a reactive hyperemia response, which is a transient surge in blood flow following cold-induced vasoconstriction [1,3,27]. This would enhance oxygen delivery upon exercise onset, possibly offsetting the initial vasoconstrictive effects of cooling and supporting higher muscular performance in the first set. The lack of a significant increase in oxygen saturation (SmO_2_) in the CWI group may indicate that oxygen utilization remained high during this set, with improved perfusion buffering early fatigue [2,18]. By contrast, the non-CWI group showed progressive increases in SmO_2_ in the gastrocnemius (after set 1) and in the soleus (after set 3), suggesting a more gradual adaptation in perfusion and oxygen delivery. This delayed response may reflect standard hemodynamic adjustments during repeated submaximal efforts in non-cooled muscle [1,21].

The myotonometry data further supports the interpretation of a temporally shifted neuromuscular activation pattern in the CWI group. In the non-CWI group, muscle tension (i.e., increased stiffness and frequency, decreased decrement and creep) changed significantly after the first set, suggesting rapid accumulation of neuromuscular fatigue [6,20]. In contrast, the CWI group exhibited significant changes primarily after the second set. These delayed neuromuscular responses may be attributed to the dampening effect of cooling on stretch reflex sensitivity and muscle spindle activity. This could reduce initial muscle tone and delay fatigue-related adaptations [4,20]. Such findings suggest that CWI acts not only as a thermal intervention but as a regulator of the timing of both vascular and neuromuscular responses. Rather than enhancing absolute capacity throughout all sets, it seems to buffer early fatigue responses, thereby redistributing work capacity across the session [5,18].

## 5. Conclusions

This study demonstrates that pre-exercise cold water immersion acutely enhances calf raise performance in resistance-trained individuals. Compared to a non-CWI condition, participants in the CWI group performed more repetitions in the first set and lifted heavier loads across sets. These effects were accompanied by altered muscle oxygenation and delayed neuromuscular responses, suggesting that CWI may optimize early performance during strength training.

## Figures and Tables

**Figure 1 jfmk-10-00328-f001:**
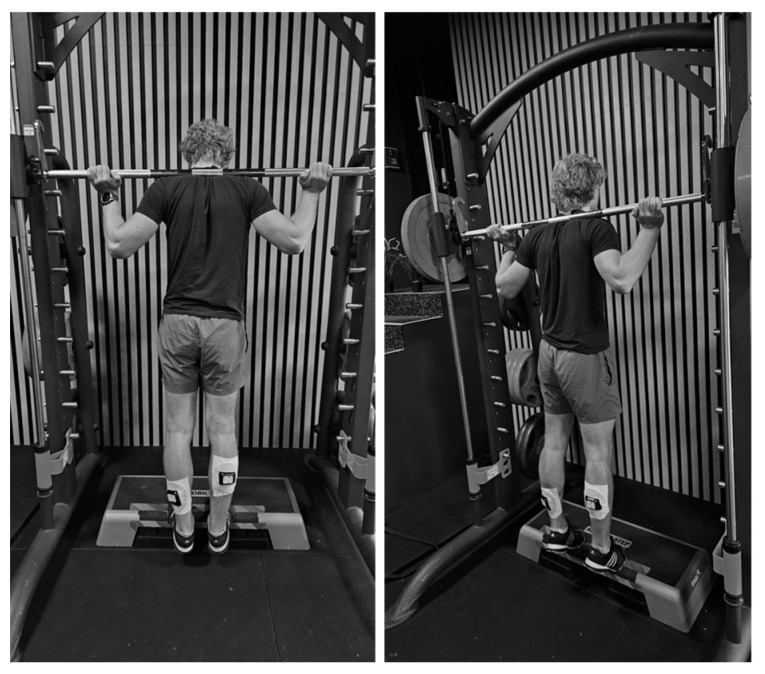
Set-up with Moxy monitor during testing calf raises in Smith machine.

**Figure 2 jfmk-10-00328-f002:**
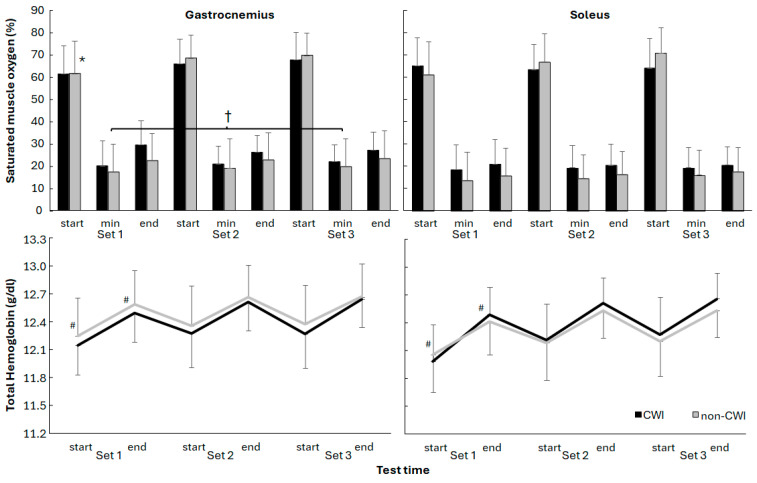
Muscle oxygen saturation and total hemoglobin of gastrocnemius and soleus over the three sets for CWI and non-CWI group. * indicates a significant difference with the other two sets for this condition. † indicates a significant difference between these two sets for this condition. # indicates a significant difference for this with the other two sets for both conditions.

**Figure 3 jfmk-10-00328-f003:**
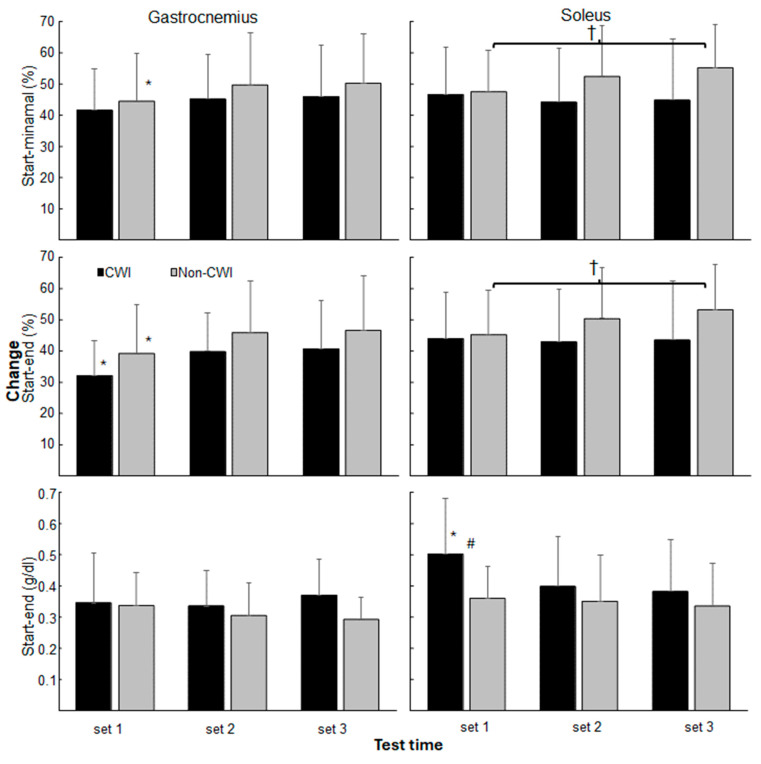
Change in muscle oxygen saturation and total hemoglobin of gastrocnemius and soleus from start to minimal and end of each set for CWI and non-CWI group. * indicates a significant difference with the other two sets for this condition. † indicates a significant difference between these two sets for this condition. # indicates a significant difference between the two groups in this set.

**Figure 4 jfmk-10-00328-f004:**
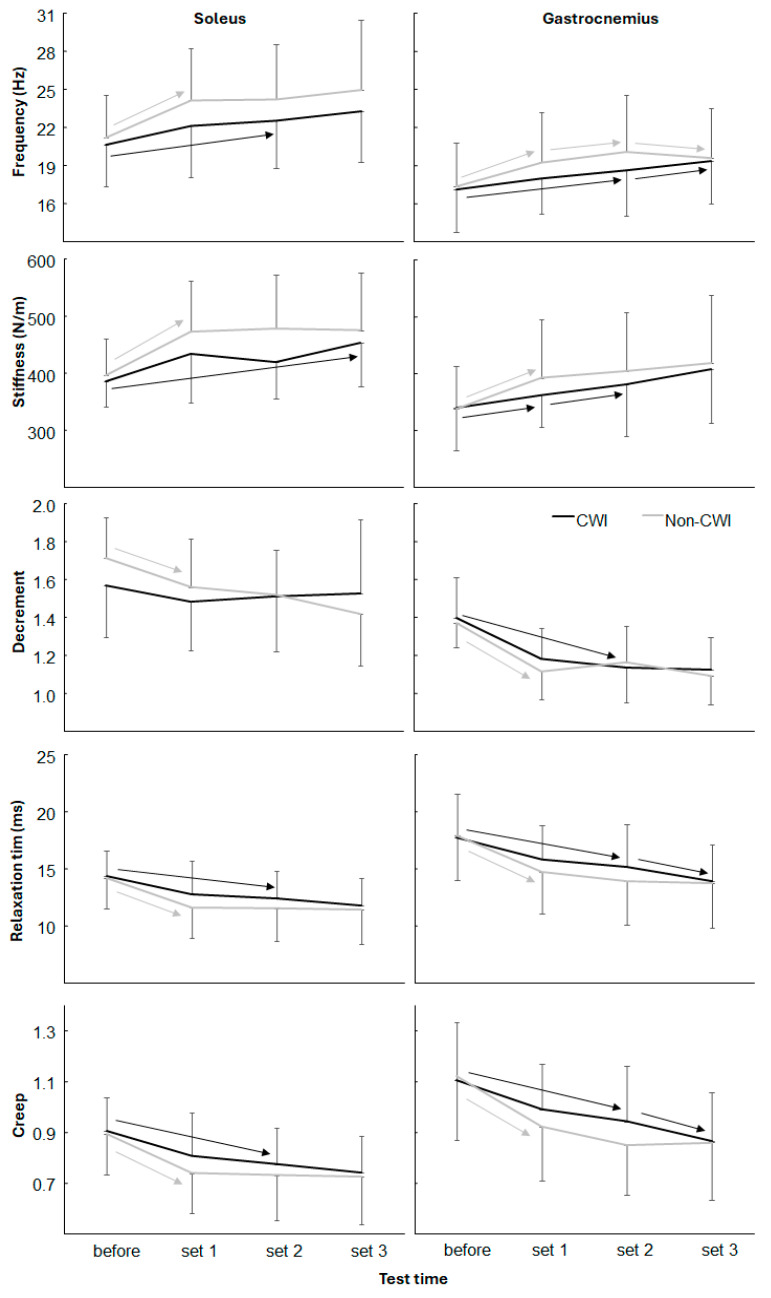
Muscle tension parameters at start and after each test for CWI and non-CWI condition. Arrow indicates a significant difference between these two test times for this condition.

**Table 1 jfmk-10-00328-t001:** Lifted load, repetitions, and rate of perceived exertion (RPE) before the first set and after each set for CWI and non-CWI group.

	Before		Set 1		Set 2		Set 3	
Condition	CWI	Non-CWI	CWI	Non-CWI	CWI	Non-CWI	CWI	Non-CWI
Load			94.5 ± 18.1	86.8 ± 26.3	98.0 ± 18.7 #	87.6 ± 26.1	97.6 ± 18.6 #	87.6 ± 26.1
Repetitions			23.7 ± 2.6 *	21.5 ± 3.3 *	16.3 ± 1.6	16.5 ± 1.6	15.6 ± 1.6	16.8 ± 3.2
RPE †	1.8 ± 1.4	2.7 ± 1.7	5.3 ± 1.6	5.8 ± 1.4	7.0 ± 1.1	7.6 ± 1.1	7.8 ± 1.4	8.2 ± 1.3

† indicates a significant increase from test to test for both conditions. * indicates a significantly higher number of repetitions than the other two sets for this condition. # indicates a significantly higher load than set 1 for this condition.

## Data Availability

The data presented in this study are available upon request from the corresponding author. The data are not publicly available due to the national laws of the Norwegian government regarding privacy.

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
