# Peer review of "The Acute Effect of Warm-Up with Cold Water Immersion upon Calf Raise Performance, Muscle Tension, and Oxygen Saturation"

_jfmk, 2025, doi:10.3390/jfmk10030328_

Round 1
Reviewer 1 Report
Comments and Suggestions for Authors
The authors investigated the acute effect of warm-up with cold water immersion upon calf raise performance, muscle tension, and oxygen saturation among resistance trained individuals. The results indicated that the pre-exercise cold water immersion may enhance acute lower-body performance. The current manuscript is well presented by the authors, and no major revisions are recommended.
authors have conducted a two-stage functional recovery program conducted by two professional tennis players. The results indicated that the program proved to be effective. The current manuscript is sound and do not recommend any significant recommendations or revisions.
Author Response
The authors investigated the acute effect of warm-up with cold water immersion upon calf raise performance, muscle tension, and oxygen saturation among resistance trained individuals. The results indicated that the pre-exercise cold water immersion may enhance acute lower-body performance. The current manuscript is well presented by the authors, and no major revisions are recommended.
Thank you
Reviewer 2 Report
Comments and Suggestions for Authors
Although the topic is timely and the experimental setup is intriguing, the manuscript suffers from several methodological flaws, interpretative issues, and presentation shortcomings that raise major concerns. In its current form, the study’s findings may mislead practitioners and applied professionals in the field, particularly given the lack of proper control conditions, underpowered design, and overinterpretation of non-significant outcomes. To ensure scientific clarity and maintain the integrity of applied exercise research, the following suggestions should be carefully addressed by the authors:
One of the primary concerns with this manuscript lies in the overall study design, which presents several critical flaws. Most notably, the use of a between-subjects design with only 12 participants per group is considerably underpowered and introduces a high degree of inter-individual variability, making it difficult to draw reliable conclusions. Although the authors reference a larger longitudinal project, there is no justification provided for the sample size used in this acute trial, nor is there an a priori power analysis to support the adequacy of the cohort.
Another important issue is the absence of a control warm-up condition. The comparison made is between cold water immersion (CWI) and no warm-up at all. This is problematic, as it reduces the clinical relevance of the findings—any acute benefit observed may simply be due to contrasting CWI with complete inactivity rather than demonstrating the specific efficacy of CWI over a standard, active warm-up. Without this comparator, the practical utility of the results is limited.
Additionally, the manuscript lacks clarity regarding the familiarization period. The authors state that participants underwent four weeks of training prior to testing, but it remains unclear whether this was supervised or whether both groups followed identical protocols. Furthermore, there is no information provided on the nature of the training, progression of load, or any other standardization procedures—details that are crucial for understanding potential adaptation effects and baseline equivalence.
The sample also shows a notable sex imbalance, with only 5 women among 24 participants. Despite this disparity, the authors do not perform subgroup analysis or discuss potential sex-related physiological differences. This omission limits the generalizability of the results and introduces bias in interpretation.
Finally, the exclusive use of the calf raise as the performance outcome is insufficiently justified. While the authors argue that this choice allows easier control and measurement, calf raises are rarely prioritized in practical resistance training or athletic performance contexts. As a result, the applicability of the findings to more commonly used and functionally relevant exercises—such as squats or deadlifts—is questionable.
Another major concern lies in the presentation of data and the statistical reporting throughout the manuscript. The reporting of results is incomplete and does not meet the standards expected for publication in a high-quality scientific journal. While F-values are mentioned for various statistical outcomes, the corresponding degrees of freedom are omitted, which limits the reader’s ability to fully interpret the findings. Additionally, the description of post hoc analyses is inconsistent and lacks essential detail, such as the specific comparisons made and whether appropriate corrections (e.g., Holm–Bonferroni) were applied uniformly. Moreover, confidence intervals are entirely absent from the results section. Their inclusion is necessary to provide a clearer sense of the variability and precision of the estimates reported.
While the authors do report partial eta-squared values for effect sizes—an appropriate metric—they fail to adequately contextualize these values. Readers are left without a clear understanding of the practical or clinical significance of the reported effects. This is compounded by a tendency to overinterpret non-significant findings. For example, statements suggesting trends (e.g., “tended to be higher”) or invoking physiological explanations (e.g., “reactive hyperemia”) are used in instances where statistical significance was not achieved. These interpretations should either be removed or explicitly labeled as speculative to maintain scientific rigor.
Furthermore, a notable omission is the absence of figures related to muscle tension data. Given the complexity of the myotonometry variables presented (e.g., frequency, stiffness, decrement), visual representations would greatly enhance transparency and aid in reader comprehension. Including clear, well-labeled graphs for these parameters is essential for ensuring the integrity and accessibility of the reported results.
Author Response
Although the topic is timely and the experimental setup is intriguing, the manuscript suffers from several methodological flaws, interpretative issues, and presentation shortcomings that raise major concerns. In its current form, the study’s findings may mislead practitioners and applied professionals in the field, particularly given the lack of proper control conditions, underpowered design, and overinterpretation of non-significant outcomes. To ensure scientific clarity and maintain the integrity of applied exercise research, the following suggestions should be carefully addressed by the authors:
One of the primary concerns with this manuscript lies in the overall study design, which presents several critical flaws. Most notably, the use of a between-subjects design with only 12 participants per group is considerably underpowered and introduces a high degree of inter-individual variability, making it difficult to draw reliable conclusions. Although the authors reference a larger longitudinal project, there is no justification provided for the sample size used in this acute trial, nor is there an a priori power analysis to support the adequacy of the cohort.
We appreciate the reviewer’s concern regarding the sample size and statistical power of our study. In response, we have now included a power analysis within the manuscript to justify the chosen sample size. Specifically, a priori power analysis conducted using G*Power (Faul et al., 2009), based on an assumed effect size of 0.93 derived from prior CWI research (Piñero et al., 2024), indicated that a total of 12 participants is sufficient to achieve a statistical power of 80% (β = 0.8) at an alpha level of 0.05. Given this, we believe that the current sample size is statistically adequate for detecting meaningful effects in the context of this acute trial.
While we acknowledge the limitations inherent in between-subject designs with relatively small samples, we note that the current design aligns with standard practice in similar acute intervention studies and is further supported by the power analysis now provided.
Another important issue is the absence of a control warm-up condition. The comparison made is between cold water immersion (CWI) and no warm-up at all. This is problematic, as it reduces the clinical relevance of the findings—any acute benefit observed may simply be due to contrasting CWI with complete inactivity rather than demonstrating the specific efficacy of CWI over a standard, active warm-up. Without this comparator, the practical utility of the results is limited.
We acknowledge the reviewer’s concern regarding the absence of a control warm-up condition. However, we respectfully disagree with the notion that the comparison between cold water immersion (CWI) and no warm-up lacks practical relevance. All participants in the current study were healthy individuals with fully functioning lower limbs and had engaged in typical daily movement (e.g., walking, stair climbing) prior to arriving at the training session. The muscles of the lower limbs—particularly the calves—are therefore unlikely to be in a fully rested state, even without a structured warm-up protocol.
Moreover, in resistance training practice, it is common for athletes to begin sessions with lighter sets of the planned exercises rather than performing a separate, structured warm-up. This type of approach is often considered sufficient by practitioners and aligns with how warm-up is typically applied in real-world strength training contexts. Given this, we believe that comparing CWI to a non-active warm-up reflects a relevant and ecologically valid contrast.
That said, we agree that including an active warm-up control condition in future research could yield valuable insights, particularly to distinguish whether any benefits of CWI are due to its cooling effects specifically, or simply the result of increased muscle activation or movement preceding exercise.
Additionally, the manuscript lacks clarity regarding the familiarization period. The authors state that participants underwent four weeks of training prior to testing, but it remains unclear whether this was supervised or whether both groups followed identical protocols. Furthermore, there is no information provided on the nature of the training, progression of load, or any other standardization procedures—details that are crucial for understanding potential adaptation effects and baseline equivalence.
We thank the reviewer for highlighting the need for greater clarity regarding the familiarization and standardization procedures. As noted in the manuscript, the four-week training period was part of a larger longitudinal study, during which all sessions were supervised by trained personnel and followed the same standardized protocol across participants. The loading progression, set structure, and execution criteria (as described in detail) were applied consistently throughout the familiarization and testing phases. We have now clarified this in the manuscript to ensure greater transparency.
The sample also shows a notable sex imbalance, with only 5 women among 24 participants. Despite this disparity, the authors do not perform subgroup analysis or discuss potential sex-related physiological differences. This omission limits the generalizability of the results and introduces bias in interpretation.
We acknowledge the sex imbalance in the sample, with only five female participants included. Although limited by sample size, we examined potential differences between male and female participants and did not observe any notable disparities between groups. Given the absence of apparent sex-related effects and the exploratory nature of this acute study, we considered it unwarranted to include further subgroup analysis or discussion. However, we agree that future studies with larger and more balanced cohorts should investigate sex-specific responses in greater detail.
Finally, the exclusive use of the calf raise as the performance outcome is insufficiently justified. While the authors argue that this choice allows easier control and measurement, calf raises are rarely prioritized in practical resistance training or athletic performance contexts. As a result, the applicability of the findings to more commonly used and functionally relevant exercises—such as squats or deadlifts—is questionable.
We acknowledge the reviewer's concern regarding the selection of the calf raise as the primary performance outcome. However, recent literature has demonstrated that muscle strength and hypertrophy adaptations in the gastrocnemius are comparable in magnitude to those observed in other major muscle groups, supporting its relevance as a target for resistance training studies. The use of the calf raise in the present study was also chosen to minimize disruption to participants' habitual training routines, thereby enhancing ecological validity and facilitating participant recruitment. Moreover, all muscle groups, including the gastrocnemius, deserve investigation in applied physiology, as each has distinct functional roles and adaptation profiles. Prioritizing only the most prominent muscle groups would unnecessarily limit our understanding of training effects across the human body.
Another major concern lies in the presentation of data and the statistical reporting throughout the manuscript. The reporting of results is incomplete and does not meet the standards expected for publication in a high-quality scientific journal. While F-values are mentioned for various statistical outcomes, the corresponding degrees of freedom are omitted, which limits the reader’s ability to fully interpret the findings. Additionally, the description of post hoc analyses is inconsistent and lacks essential detail, such as the specific comparisons made and whether appropriate corrections (e.g., Holm–Bonferroni) were applied uniformly. Moreover, confidence intervals are entirely absent from the results section. Their inclusion is necessary to provide a clearer sense of the variability and precision of the estimates reported.
We have included the degrees of freedom in the text now. Holm-bonferoni corrections are uniformly used. However, when effects between groups are compared this is only on two levels and thereby no correction is necessary. In JASP this is all automatically calculated. We have written many articles before with the same used statistics and it is very clear that we always use a Holm-bonferoni correction when necessary. It is even NOT possible, when you have a variable on more then 3 levels to perform, in JASP, a post hoc correction without correction.
In our opinion it is normal to report in Figures data with a mean and standard deviation as it indicates the variability around the average. One standard deviation, which is used in the figures shows around 68% of variance, while two standard deviations shows the variability of 95% of variance which is almost similar to the confidence intervals. Confidence intervals are also calculated based upon standard deviation and sample means. As they are very related to each other and we present mainly figures, we think it is better to use standard deviations and not confidence intervals as this is regular praxis in figures. We hope the reviewer agrees wit this.
While the authors do report partial eta-squared values for effect sizes—an appropriate metric—they fail to adequately contextualize these values. Readers are left without a clear understanding of the practical or clinical significance of the reported effects. This is compounded by a tendency to overinterpret non-significant findings. For example, statements suggesting trends (e.g., “tended to be higher”) or invoking physiological explanations (e.g., “reactive hyperemia”) are used in instances where statistical significance was not achieved. These interpretations should either be removed or explicitly labeled as speculative to maintain scientific rigor.
We agree with the reviewer and have rewritten those part to avoid confusion for the readers. Difference in saturation in soleus muscle (reactive hyperemia) was shown to be significantly different as shown in figure 3. We have now referred to that.
Furthermore, a notable omission is the absence of figures related to muscle tension data. Given the complexity of the myotonometry variables presented (e.g., frequency, stiffness, decrement), visual representations would greatly enhance transparency and aid in reader comprehension. Including clear, well-labeled graphs for these parameters is essential for ensuring the integrity and accessibility of the reported results.
In figure 4 we have shown all the different parameters of muscle tension, so we don’t understand what is missing of figures.
Reviewer 3 Report
Comments and Suggestions for Authors
Thank you for the opportunity to review the article ‘The acute effect of warm-up with cold water immersion upon 2 calf raise performance, muscle tension and oxygen saturation.’
This is a relevant topic in the field of musculoskeletal health, especially with regard to water immersion.
To make it easier to review, I will do so in sections, making it easier for authors to read.
Abstract
The abstract does not provide information on the type of study that was conducted.
The results section does not present values, which makes it difficult to read.
The conclusion does not address the proposed objectives.
Regarding keywords, I recommend that acronyms not be used.
Introduction
The introduction presents all the topics to be developed, although they do not seem to be well organised in terms of readability. It would be beneficial to rewrite the introduction to make it more comprehensible and fluid to read.
I believe that the objective should be clearer and essentially operational, because as it is currently governed, this objectivity is not apparent.
Materials and Methods
How were participants recruited? This is not described in the study.
Who assessed the participants? Were they healthcare professionals?
There is no reference to how the groups were randomised. Or was it not done? Can you clarify this point?
Were participants blinded to the group allocation? What about the researchers?
This information should be clarified in the paper.
It is not very clear how the load progression was carried out. Could you clarify this part a little better?
Results
This section should be rewritten, as what is in the text is actually in the table or figure. In other words, the information is repeated twice instead of presenting only the most relevant information in the text.
Discussion
Well-organised discussion, although very long sentences are used. It would be advisable to divide the paragraphs further in order to improve coherence in the reading and approach to the topics and their discussion.
Some statements do not include bibliographical references and should be reviewed.
Conclusion
This conclusion already addresses the objectives and is more organised than the one in the abstract, but it is too long. It should only address the objectives.
Author Response
We have answered to all comments of the reviewer and made the changes in the manuscript in red. We think that it is now suitable for publication.
Reviewer 3
Thank you for the opportunity to review the article ‘The acute effect of warm-up with cold water immersion upon 2 calf raise performance, muscle tension and oxygen saturation.’
This is a relevant topic in the field of musculoskeletal health, especially with regard to water immersion.
To make it easier to review, I will do so in sections, making it easier for authors to read.
Abstract
The abstract does not provide information on the type of study that was conducted.
The results section does not present values, which makes it difficult to read.
The conclusion does not address the proposed objectives.
Regarding keywords, I recommend that acronyms not be used.
We have made changes to these sections now.
Introduction
The introduction presents all the topics to be developed, although they do not seem to be well organised in terms of readability. It would be beneficial to rewrite the introduction to make it more comprehensible and fluid to read.
I believe that the objective should be clearer and essentially operational, because as it is currently governed, this objectivity is not apparent.
Thank you for your constructive feedback regarding the clarity and organization of the introduction. Based on your suggestions, the introduction has been thoroughly revised to improve its logical structure, readability, and flow. Efforts were made to create a clearer narrative progression from general background to the identification of research gaps, with seamless transitions between topics. In particular, the purpose of the study has now been more clearly articulated to enhance the operational clarity of the manuscript. All references remain unchanged, and their integration has been preserved within restructured sentences to ensure scientific accuracy and continuity.
Materials and Methods
How were participants recruited? This is not described in the study.
Who assessed the participants? Were they healthcare professionals?
There is no reference to how the groups were randomised. Or was it not done? Can you clarify this point?
Were participants blinded to the group allocation? What about the researchers?
This information should be clarified in the paper.
It is not very clear how the load progression was carried out. Could you clarify this part a little better?
Participants were recruited through personal networks at the university and via posters placed on campus. Group allocation was performed using an online randomization tool (Randomlists, n.d.). Although neither the participants nor the researchers were blinded to group allocation, all training sessions were supervised by the same individuals. To minimize bias and ensure consistency, standardized protocols were applied for both load progression and training to failure criteria. This is now mentioned in the text.
Regarding the load progression method, this was described in the manuscript as follows:
“The prescribed load was based on the number of repetitions completed in the previous session. The load that allowed participants to perform 15–20 repetitions in the last set of the previous session was used as the starting load. If participants completed 21–24 repetitions, the load was increased by 2.5 kg in the following session. If more than 25 repetitions were achieved, the load was increased by 5 kg. If fewer than 15 repetitions were completed, the load was reduced by 2.5 kg in the next session.”
We believe this approach provides a clear and systematic progression model, allowing for individualized load adjustments based on actual performance while maintaining training intensity within the target repetition range.
Results
This section should be rewritten, as what is in the text is actually in the table or figure. In other words, the information is repeated twice instead of presenting only the most relevant information in the text.
In our opinion it is normal to write I the text what is found and then show it in the figure as we have done. It is important to show the F, p and effect size values in the text and then show in figures the development. This is normal praxis for most articles we have written and not called double presentation as that is according to us when you show the same results in both a figure and a table. Now readers can read the statistics in the text and the quick overview in the figures and tables. We hope that the reviewer agrees with our point of view.
Discussion
Well-organised discussion, although very long sentences are used. It would be advisable to divide the paragraphs further in order to improve coherence in the reading and approach to the topics and their discussion.
Some statements do not include bibliographical references and should be reviewed.
Based on your suggestion, I have revised the discussion section by shortening and simplifying several long sentences to improve clarity and readability. However, I chose not to divide the paragraphs further, as doing so disrupted the logical flow and coherence of the discussion. I believe the current paragraph structure maintains a better continuity in the development of ideas while still being easier to follow after the sentence-level edits.
Conclusion
This conclusion already addresses the objectives and is more organised than the one in the abstract, but it is too long. It should only address the objectives.
In response, the conclusion has been revised to focus more directly on the study’s objective, removing extended interpretation and limiting the content to the key findings. The new version highlights only the acute performance effects of pre-exercise cold water immersion on resistance training, aligning more clearly with the study’s aim and improving conciseness and clarity.
Round 2
Reviewer 3 Report
Comments and Suggestions for Authors
Thank you for the opportunity to review this paper again.
The improvements made after the review were fundamental. The paper is now clearer, more objective and better organised.
I just have one comment regarding the results.
Thank you for your clarification regarding the presentation of results. I understand your point of view and the rationale behind including both the statistical details in the text and the visual representation in the figures/tables. While my initial concern was to avoid what could be perceived as redundancy, I acknowledge your explanation and will leave this aspect to the discretion of the editorial team.